# Ergosterol Is Critical for Sporogenesis in *Cryptococcus neoformans*

**DOI:** 10.3390/jof10020106

**Published:** 2024-01-26

**Authors:** Amber R. Matha, Xiaofeng Xie, Xiaorong Lin

**Affiliations:** Department of Microbiology, University of Georgia, Athens, GA 30602, USA

**Keywords:** *Cryptococcus neoformans*, mating, sporulation, ergosterol

## Abstract

Microbes, both bacteria and fungi, produce spores to survive stressful conditions. Spores produced by the environmental fungal pathogen *Cryptococcus neoformans* serve as both surviving and infectious propagules. Because of their importance in disease transmission and pathogenesis, factors necessary for cryptococcal spore germination are being actively investigated. However, little is known about nutrients critical for sporogenesis in this pathogen. Here, we found that ergosterol, the main sterol in fungal membranes, is enriched in spores relative to yeasts and hyphae. In *C. neoformans*, the ergosterol biosynthesis pathway (EBP) is upregulated by the transcription factor Sre1 in response to conditions that demand elevated ergosterol biosynthesis. Although the deletion of *SRE1* enhances the production of mating hyphae, the *sre1*Δ strain is deficient at producing spores even when crossed with a wild-type partner. We found that the defect of the *sre1*Δ strain is specific to sporogenesis, not meiosis or basidium maturation preceding sporulation. Consistent with the idea that sporulation demands heightened ergosterol biosynthesis, EBP mutants are also defective in sporulation. We discovered that the overexpression of some EBP genes can largely rescue the sporulation defect of the *sre1*Δ strain. Collectively, we demonstrate that ergosterol is a critical component in cryptococcal preparation for sporulation.

## 1. Introduction

Sporulation is a cell survival strategy employed by prokaryotic and eukaryotic microbes to survive in harsh conditions that, otherwise, inhibit vegetative growth. Bacteria, such as *Bacillus*, can produce endospores via asymmetric cell division, which results in the production of a spore developing inside the mother cell [1]. These spores are resistant to various types of stress that the mother cells are sensitive to, allowing the spores to persist [2].

One defining feature distinguishing eukaryotes from prokaryotes is that spores can be produced in the former organisms via sexual reproduction with meiosis. One of the model species for studying sexual reproduction is the pathogenic basidiomycete fungus *Cryptococcus neoformans*. This fungus was first characterized in 1894 [3] and has been recognized as an asexual yeast with mitotic divisions both in the environment and the host. The sexual cycle of *C. neoformans,* with its characteristic morphological changes from yeasts to hyphae and to fruiting bodies, was not defined until the 1970s by Dr. June-Kwon Chung [4,5]. In addition to producing resilient sexual spores, the morphological transition to hyphae better allows for nutrient scavenging and protection against predation by soil amoeba [6,7].

Given the fact that spores are resistant to stressful environments and are infectious agents that can cause fatal cryptococcal meningitis [8,9], several groups, including the pioneers Dr. June-Kwon Chung and Dr. Joseph Heitman, have dedicated their work to dissecting the mating process [5,10,11,12,13,14,15]. Two modes of sexual reproduction are discovered in this fungus: unisexual and bisexual reproduction [4,16]. Unisexual reproduction is mainly achieved via cells of a single mating type generating hyphae and spores after endoduplication to double its DNA content prior to meiosis [14]. By contrast, bisexual mating occurs when cells of two mating types, α and a, conjugate with each other in response to pheromones, forming a dumbbell-shaped zygote [13] (Figure 1A). The zygote then grows polarly as dikaryotic hyphae with two parental nuclei migrating in synchronization without nuclear fusion. Hyphal extension can occur indefinitely until unknown stimuli cause the aerial hyphal tip to swell and become a basidium. Here, the two parental nuclei fuse, undergoing meiosis and repeated rounds of mitosis to give rise to four haploid spore chains that bud off the surface of the basidium (Figure 1A) [4,10,17].

Spores are markedly reduced in metabolic activities compared to other cell types [8,18,19]. Because of this, spores must come packaged with much of the cellular material necessary for germination, since the generation of major biomolecules requires the transcription and translation of the enzymes to build them. Ergosterol is a major component of the cell membrane, responsible for membrane rigidity and integrity. Current antifungal drugs used to treat cryptococcosis, such as amphotericin B or fluconazole, target ergosterol directly or the EBP pathway. Consequently, the EBP mutants show increased sensitivity to these antifungals [20,21]. Spores are particularly resistant to antifungal drug exposure [22]. Thus, we hypothesize that the EBP pathway is important in sexual reproduction, especially sporulation.

The main transcription factor that upregulates the transcription of EBP genes is Sre1. Previously, Sre1 or its homologues have primarily been investigated for their role in response to hypoxia in humans and fungi [23,24]. Little research has been conducted to investigate other roles this transcription factor might have during normal cellular development. Given the critical nature of cholesterol for oogenesis in mammals [25,26,27], we postulate that ergosterol is important during sexual reproduction in lower eukaryotes like fungi.

Analyzing recently published RNA-seq data indicates that *SRE1* is upregulated (almost four-fold increase) on mating-inducing V8 media compared to rich YPD media favorable for vegetative yeast growth [28]. The transcript level of *SRE1* is further increased over time on V8 media [29], suggesting that this transcription factor might be more critical for the later step(s) of the mating process. Surprisingly, a previous study reported that the *sre1*Δ mutant showed enhanced mating based on visual observation of increased filamentation [30]. However, the potential role of Sre1 in the later steps of the mating process has not been thoroughly investigated. In this study, we found that although the *sre1*Δ mutant indeed showed enhanced filamentation when crossed with a wild-type partner compared to a wild-type cross during bisexual mating, the mutant cross was deficient at sporulation. As we demonstrate in this paper, this defect is tightly linked to the role of Sre1 in increasing EBP genes.

## 2. Materials and Methods

### 2.1. Strains and Growth Conditions

The *C. neoformans* strains used in this study are listed in Appendix A. Strains were stored at −80 °C in glycerol stocks and freshly streaked out prior to experimentation. Cells were maintained on YPD medium at 30 °C unless stated otherwise.

### 2.2. Mating

Mating was conducted on V8 pH = 5 agar media for serotype A crosses and V8 pH = 7 agar media for serotype D crosses [11]. Equal number of cells of both mating partners (OD_600_ = 3) were mixed together, and 10 µL of this mixture was spotted onto the plate. The mating crosses were carried out at 21 °C in the dark for approximately two weeks.

### 2.3. Growth Assays

To examine vegetative yeast growth, the tested strains were grown in a YPD liquid medium overnight at 30 °C with shaking. Cells were washed with sterile water and adjusted to a cell density of OD_600_ = 0.1. The cell suspensions were then inoculated into a 96-well plate incubated at 30 °C for two days in an Epoch 2 plate reader. OD_600_ was read every hour for the duration of the experiment. 

To examine fluconazole sensitivity, the tested strains were grown in YPD liquid medium overnight at 30 °C with shaking. Cells were washed with sterile water, adjusted to a cell density of OD_600_ = 1, and serially diluted in 10-fold increments. The prepared cells were spotted onto YNB agar media as well as YNB agar media with the indicated concentration of fluconazole.

### 2.4. Gene Manipulation

For gene deletion, the *SRE1* deletion cassette was amplified from genomic DNA (gDNA) of a *sre1*Δ mutant, which is part of the *C. neoformans* transcription factor deletion library generated by Dr. Yong-Sun Bahn and colleagues [30]. All primers used in this study are listed in Appendix A. To generate the sgRNA for the *SRE1* deletion, the *U6* promoter and sgRNA scaffold were amplified from JEC21 gDNA and the plasmid pDD162, using primer pairs Linlab4627/Linlab7751 and Linlab4628/Linlab7752, respectively. The *U6* promoter and sgRNA scaffold pieces were fused together by overlap PCR with primers Linlab4594/Linlab4595 to generate the final sgRNA construct as described previously [31,32].

For gene overexpression, open reading frames (ORFs) were amplified via PCR using the genomic DNA of *C. neoformans* H99 as the template and cloned into vectors containing either the *TEF1* (pLinlab995) or the *GPD1* (pLinlab1059) promoter. The resulting plasmids were confirmed via restriction enzyme digestion. M13F and M13R primers were used to amplify the donor DNA for Transient CRISPR-Cas9 coupled with electroporation (TRACE) [32], and constructs were inserted into the safe haven *SH2* region [33].

Overexpression and deletion constructs were transformed into the indicated *C. neoformans* strains via TRACE [31,32]. Transformants were selected on YPD medium with 100 μg/mL of nourseothricin (NAT), 100 μg/mL of neomycin (NEO), or 200 μg/mL of hygromycin (HYG), depending on the drug marker used. 

The successful deletion of *SRE1* was screened via diagnostic PCR. Primers 8488 and 4897, which lie on the *SRE1* open reading frame, were used to ensure that the ORF was missing. Primer pairs 4895 (a *SRE1* promoter forward) and 3792 (an inside NAT reverse primer) were used to ensure that the NAT cassette was inserted into the *SRE1* original locus. The successful integration of EBP gene overexpression constructs into the *SH2* region was screened via 3-primer PCR [34]. Since the construct can be inserted into the *SH2* region in either the forward or the reverse direction, primer 5936, a reverse primer on the overexpression construct, was paired with *SH2* sequencing primers 4814 and 4815. Regardless of the direction in which the construct was inserted into the *SH2* site, or whether it was not inserted at all, a band would amplify with the three primer PCRs, and the size of the band indicated which of the three situations had occurred in the tested transformants.

All primers and plasmids used in this study are listed in Appendix A.

### 2.5. RNA Extraction and Real-Time PCR

Real Time PCR (RT-PCR) was used to confirm the ergosterol biosynthesis gene overexpression. Cells were grown in YPD liquid cultures with shaking at 30 °C overnight. They were collected, flash-frozen with liquid nitrogen, and lyophilized overnight. Desiccated cells were manually disrupted with glass beads, and total RNA was extracted using the PureLink RNA Mini Kit (Invitrogen, Waltham, MA, USA) according to the instruction of the manufacturer. To remove DNA contamination, samples were treated with DNase using the TURBO DNA-free Kit (Invitrogen). First strand cDNA was synthesized using the GoScript Reverse Transcription System (Promega, Madison, WI, USA) following the manufacturer’s instructions. Power SYBR Green (Invitrogen) was used for all RT-PCR reactions. *TEF1* was used as an endogenous control for all RNA samples. All RT-PCR primers used are listed in Appendix A. Relative transcript level was determined using the ΔΔCt method as we described previously [35], and statistical significance was determined using Student’s *t*-test.

### 2.6. Microscopy

For fluorescence observation of filipin staining, or Dmc1-mCherry, samples were observed with a Zeiss Imager M2 microscope equipped with an AxioCam 506 mono camera. Filipin was visualized with the FL Filter Set 49 DAPI (Carl Zeiss Microscopy, Munich, Germany). The mCherry signal was visualized with the FL filter set 43 HE cy3 (Carl Zeiss Microscopy). All fluorescence images except field of view images were taken with a Zeiss 63x apochromat oil objective with a numerical aperture of 1.4. The field-of-view images were taken with the Zeiss 40x Plan-Neofluar objective lens with a numerical aperture of 0.75.

The fluorescence intensity, cell length, and cell diameter were quantified via Zen Pro software (Carl Zeiss Microscopy). Yeast and basidial surface areas were estimated by calculating the surface area of a sphere:
(1)
$${SA}_{yeast/basidium}=4\pi r^{2}$$


Hyphal and spore surface areas were estimated by calculating the surface area of a cylinder:
(2)
$${SA}_{hypha/spore}=2\pi r(r+h)$$


For mating colony imaging, a SZX16 stereoscope (Olympus) was used to observe whole colony and colony edge morphologies. Images were captured with an AxioCam 305 camera (Carl Zeiss Microscopy) and acquired using the Zen Pro software (Carl Zeiss Microscopy).

For imaging of basidia still on the V8 agar, a CX41 light microscope (Olympus) equipped with an AxioCam 305 camera (Carl Zeiss Microscopy) was used. The objective used for these images was an Olympus PlanC N 10X objective with a numerical aperture of 0.25.

### 2.7. Filipin Staining

Equal numbers of cells of both mating partners were mixed and cocultured on V8 pH = 5 plates and incubated at 21 °C in the dark for two weeks. Spores and basidia were scraped from the agar and inoculated into dH_2_O. Then, 1 µL of 50 mg/mL filipin dissolved in DMSO was added to 99 µL of cell suspension. After 15 min of incubation in the dark, the cells were visualized. Fluorescence intensity was quantified in Zen Pro software (Carl Zeiss Microscopy).

## 3. Results

### 3.1. Ergosterol Is Enriched in Basidia and Spores Based on Filipin Staining

Cryptococcal spores contain fewer ergosterol biosynthesis pathway proteins (EBPs) than yeast cells [18]. In *Fusarium* spp., EBP proteins increase only after 7–11 h post germination [36]. Therefore, spores must utilize the already present ergosterol, either stored in lipid bodies or the membrane that is derived from basidial cell membrane, for renewed growth before sufficient new ergosterol can be synthesized. We hypothesized that ergosterol would be enriched in basidia (also known as stem cells for sporulation) and spores. To test our hypothesis, we used the filipin stain to quantify relative ergosterol levels in different cryptococcal cell types. Filipin specifically binds to cholesterol in animals and ergosterol in fungi [37]. We found that the median filipin signal was 40-fold higher in spores than that in yeast cells or hyphae per surface area (Figure 1B,C). The median filipin signal for basidia was five times higher than yeast cells (Figure 1B,C). We noticed that ergosterol was not even distributed in the cell membrane. Rather, it was accumulated at the leading edge of the basidial heads and at the septa of hyphae. The distribution of filipin stain was more even along the periphery of spores and yeast cells.

### 3.2. Crosses Involving sre1Δ Produce Abundant Hyphae but Are Defective at Sporulation

Ergosterol is synthesized from acetyl-CoA through 19 reactions carried out by 23 EBP enzymes. Sre1 is a transcription factor that governs the transcription of EBP genes in response to conditions that require enhanced ergosterol biosynthetic activities such as hypoxia or in the presence of antifungals that inhibit EBP enzymes [23,38]. Although deletion of the *SRE1* gene reduces the total amount of ergosterol in the cell [23] based on filipin staining (Figure 2A), vegetative yeast growth of the *sre1*Δ mutant is comparable to the wild type (Figure 2B).

During bisexual reproduction, two yeast cells of opposite mating types fuse and form dikaryotic hyphae. Given that H99 background strains do not self-filament on V8 media, filamentation is only produced after successful bisexual mating, which is visually observable as a fluffy white edge from the original yeast colonies with mixed **a** and α cells (Figure 2C). Here, we found that in unilateral bisexual crosses where the *sre1*Δ mutant mated with a compatible wild-type partner (*sre1*Δα xH99α or *sre1*Δα x KN99**a**), more abundant filamentation was produced compared to the WT cross (Figure 2C). This is in agreement with previous observations that describe this mutant as being enhanced for mating [30]. Interestingly, a bilateral cross between two *sre1*Δ mutants had reduced filamentation (Figure 2C). This suggests that one copy of *SRE1* is necessary during bisexual mating to cause the enhanced filamentation phenotype.

Basidia and basidiospores form after two weeks in a wild-type cross on V8 media. More than 80% of basidial heads carried four spore chains by this time point, resulting from repeated mitosis after one meiotic event (Figure 1A and Figure 2C). However, in unilateral crosses where one of the mating partners was a *sre1*Δ mutant (e.g., α x *sre1*Δ**a** or **a** x *sre1*Δα), 92% of the basidia were deficient at producing spores after two weeks (Figure 2C). The sporulation defect persisted even after three and four weeks. There were rare occurrences of spore chains forming in *sre1*Δ unilateral crosses (8% of basidia); however, the vast majority of the basidia observed were barren (no spores) or had abnormal numbers of single spores (Figure 2D). Only ~10% of basidia were observed producing spore chains in unilateral crosses involving a *sre1*Δ mutant, compared to ~80% of basidia in the wild-type cross.

The fact that this sporulation defect was observed when the *SRE1* gene was deleted in either the **a** or the α mating partner indicates that this is not a mating type-specific phenomenon, and there is a haploinsufficiency for Sre1 during sporulation. Thus, functional Sre1 from both mating partners is required for normal sporulation. The bilateral cross of two *sre1*Δ partners (*sre1*Δ**a** x *sre1*Δα) showed an even more severe defect in sporulation where 95% of basidia were barren, 5% showed single spores, and none carried spore chains (Figure 2D). To test if the role of Sre1 in sporulation is specific to the serotype A reference strain H99, we deleted the *SRE1* gene in the serotype D congenic pair strains JEC21α/JEC20**a**. The serotype D reference pair strains mate more robustly than the serotype A reference strain pair H99α/KN99**a**. Interestingly, a unilateral cross between wildtype JEC20**a** and the *sre1*Δ JEC21α did not filament as well as the wild type JEC21α/JEC20**a** cross (Appendix A). Nonetheless, the unilateral cross involving a *sre1*Δ mutant showed a sporulation defect, with 92% of basidia being barren or having produced only a single spore (Appendix A). Therefore, the impact of haploid insufficiency of *SRE1* on sporulation is conserved among the *C. neoformans* species.

It is known that maturation of basidia plays an important role in the ability of the basidial head to produce spores [29]. Thus, we hypothesized that the defect in sporulation could be caused by a defect in the basidial head maturation process. The basidial maturation score (BMS), an index derived from the division of the diameter of the widest part of the basidium by the diameter of the hypha at the base of the basidium (Figure 2E), was recently developed to determine the maturity of the basidia [29]. Basidia with maturation scores greater than 1.6 are considered mature basidia [29]. We measured the diameter of basidia generated from a wild-type cross, a *sre1*Δ unilateral cross, and a *sre1*Δ bilateral cross after 2 weeks. We found that all H99 BMS scores were greater than 1.6, indicating all of them were mature by this time point. Although the *sre1*Δ unilateral cross had some basidia of low BMS, 75% of the basidia examined were mature basidia that should be capable of sporulation (Figure 2E). Likewise, 71% of basidia from the *sre1*Δ bilateral cross were also mature (Figure 2E). Basidia from both *sre1*Δ unilateral and bilateral crosses had single spores protruding with BMS scores at or over 1.6, similar to the wild-type basidia. Therefore, the minor population of immature basidia in crosses involving the *sre1*Δ mutant does not account for the sporulation defect observed in the vast majority of the basidia.

### 3.3. sre1Δ Sporulation Defect Is Not Due to a Defect in Meiosis Gene Activation

The basidial head is the site of karyogamy, meiosis, and sporogenesis during bisexual reproduction [10] (Figure 1A). Mature basidia from the *sre1*Δ mutant that were incapable of producing spores could be deficient in completing meiosis or sporogenesis. Because cholesterol intermediates have been identified as activators of meiosis in oocytes in vitro [33], we hypothesized that meiosis may be blocked in the *sre1*Δ mutant due to reduced sterol content. Dmc1 is a recombinase known to mediate homologous recombination during double-stranded break repair caused by crossing-over events in meiosis. In *C. neoformans*, Dmc1 is required for meiosis and sporulation [14]. Dmc1 is exclusively expressed in aerial hypha tips as the basidium develops and it disappears once sporulation starts [39]. The expression of this gene has also been used to determine if genes of interest are up or downstream of the meiotic event during mating [29].

To examine meiotic process, we decided to monitor the activation of Dmc1 using a fluorescently labeled Dmc1 with its expression driven by its own promoter. We found that Dmc1 was expressed in basidia generated from a *sre1*Δ unilateral bisexual cross, similar to its expression in basidia produced by a wild-type cross (Figure 3A). Thus, meiosis was initiated in the basidia from the *sre1*Δ involved crosses. The fluorescence intensity of the basidia had no significant difference, suggesting that the amount of Dmc1 was comparable between both crosses (Figure 3B). As four meiotic nuclei are generated as the end product of meiosis, we decided to determine if meiosis can be completed in the *sre1*Δ involved crosses by examining the presence of nuclei in mature basidia. To visualize the nuclei, we used a strain constitutively expressing mNeonGreen with a nuclear localization signal (NLS-mNG) as the mating partner of the *sre1*Δ mutant [40]. We found basidia with more than two nuclei, suggesting that meiosis was completed in these crosses (Figure 3C). Taken together, our results suggest that basidial maturation and meiosis can be completed in the *sre1*Δ involved crosses, and the defect of sporulation is most likely due to a defect in the late sporogenesis process itself.

### 3.4. Defective EBP Pathway Leads to Sporulation Defect

The EBP consists of 23 proteins that execute a series of reactions to transform acetyl Co-A into ergosterol (Figure 5A). Many of these genes are essential, but a few genes (*ERG*2, 3, 4, and 5) encoding enzymes that function in the late step reactions are not ([41], Figure 5A). To determine if mutants defective in the EBP pathway enzymes are also defective in sporulation, we examined filamentation and sporulation of crosses involving *erg3*Δ and *erg4*Δ mutants (Figure 4). Filipin staining showed that the *erg4*Δ and the *erg3*Δ mutants were less ergosterol rich than the H99 wildtype (Figure 4A). Indeed, the filipin signal in wildtype cells was 1.8 and 2.1 times higher than *erg4*Δ and *erg3*Δ, respectively (Figure 4B). This reduction in filipin signal is similar to the reduction caused by the *sre1*Δ (Figure 2A).

Unilateral crosses with these mutants generated abundant hyphae (mutant x wild type), but filamentation was severely impaired in bilateral crosses (mutant x mutant) (Figure 4C,D). This indicates that one copy of the penultimate EBP genes is sufficient for filamentation to occur but is not sufficient to support wildtype levels of sporulation. Indeed, filamentation did not occur in a cross where neither parent possesses these EBP genes. This shows that the final ergosterol product is necessary for successful filamentation as well as sporulation. For instance, 55% of basidia from an *erg4*Δ unilateral cross were barren, and 16% bore single spores. Only 29% carried four chains of spores (Figure 4E,F). The *erg3*Δ unilateral cross yielded similar sporulation frequencies with 45% of basidia being barren, 16% having single spores, and 28% of basidia producing four chains of spores. The results indicate that sporulation is sensitive to even a modest deficiency of sterols, mimicking the unilateral crosses of the *sre1*Δ mutant.

### 3.5. Overexpression of Multiple Individual EBP Genes Partially Restore sre1Δ’s Sporulation Defect

As Sre1 upregulates the expression of some EBP genes in response to stresses that require increased expression of the EBP pathway (Figure 5A), we wondered if overexpressing EBP genes could compensate for the lack of Sre1 and rescue the sporulation defect. Here, we selected *ERG2*, *ERG11*, *ERG25*, *ERG26*, and *ERG27* for overexpression. Erg25 was chosen because it is a known direct target of Sre1 [24]. Erg26 and Erg27 are known to form complexes with Erg25 in *Saccharomyces* [41,42]. We chose Erg11 because it is the target of fluconazole and Erg2 because it is a target of the morpholine class of antifungal drugs and lies downstream of *ERG11* [43]. The overexpression constructs of these *ERG* genes driven by the *TEF1* promoter were all integrated into the “safe haven” region *SH2* of the H99 genome to avoid any position effect [33].

The overexpression of these *ERG* genes was confirmed via RT-PCR (Appendix A). The overexpression of *ERG2*, *ERG11*, *ERG25*, *ERG26*, and *ERG27* genes in wild-type H99 background increased cryptococcal resistance to fluconazole (Figure 5B), indicating that these overexpressed enzymes are functional. Notably, all the ERG genes selected for overexpression encode enzymes that function downstream of Erg11, the target of fluconazole. Thus, it appears that increased expression of any of the single genes in the EBP downstream of Erg11 is sufficient to overcome Erg11 inhibition by fluconazole.

The *sre1*Δ mutant, as expected, was susceptible to fluconazole (Figure 5C). In the mutant background, of the five *ERG* genes tested, only overexpression of *ERG11* increased its tolerance to fluconazole. Therefore, the drug’s target, Erg11, can confer fluconazole tolerance to the *sre1*Δ mutant, but the effect of overexpression of any other individual EBP gene cannot overcome fluconazole sensitivity caused by *SRE1* deletion.

With the confirmation of the functionality of the overexpressed *ERG* genes, we proceeded to examine if the overexpression of these genes could rescue *sre1*Δ’s sporulation defect. The overexpression of any of the five *ERG* genes in the wild-type H99 background did not alter sporulation (Figure 5D,E). Interestingly, the overexpression of *ERG2, ERG11, ERG25*, and *ERG26* partially rescued the sporulation defect of *sre1*Δ in unilateral crosses (Figure 5D,E). Whereas the *sre1*Δ unilateral cross only produced spore chains ~10% of the time, the *ERG2*^OE^*sre1*Δ unilateral cross produced spore chains ~50% of the time, and the *ERG11*^OE^*sre1*Δ produced spore chains ~60% of the time. The *ERG25*^OE^*sre1*Δ and *ERG26*^OE^*sre1*Δ unilateral crosses produced spore chains ~70% of the time (Figure 5E). The additive effect of EBP gene overexpression plus one wild-type copy of *SRE1* in these unilateral crosses (*sre1*Δ x WT) may explain why the sporulation defect was partially rescued during unilateral bisexual mating, but the vegetative growth of these EBP gene overexpression in the haploid *sre1*Δ mutant was not restored on YNB+fluconazole. Taken together, we concluded that ergosterol is a critical nutrient for sporogenesis, and sporogenesis demands heightened expression of EBP genes.

## 4. Discussion

Cholesterol is an important component of meiosis and oocyte development in mammals and the nematode *Caenorhabditis elegans* [44]. Fungi undergo a sexual cycle that results in the production of meiotic spores. Here, we found that ergosterol is enriched in basidia and spores of the fungus *C. neoformans*, and, thus, we sought to determine if this fungal sterol was critical for sexual reproduction in *C. neoformans*. Indeed, we found that genetic mutations in the EBP pathway, such as deletion of *ERG3* and *ERG4* gene, compromise sporulation even in unilateral crosses where only one mating partner is deficient in the EBP pathway (haploid insufficiency). The importance of EBP genes in cryptococcal sporulation is also in agreement with previous studies on *Aspergillus fumigatus* where *ERG4* and *ERG5* are required for conidiation [45,46]. *ERG3* mutants do not have a conidiation defect in *A. fumigatus* [47], but conidiation defects of this mutant have been reported in *Fusarium oxysporum* [48]. Thus, the *ERG4* and *ERG3* phenotypes are conserved across phyla.

Modulating ergosterol content via the overexpression or reducing expression of EBP genes in *Aspergillus oryzae* reduces sporulation rates. Specifically, *ERG19* overexpression and RNAi knockdown both caused the number of spores produced by *A. oryzae* to decrease dramatically [49]. Both gene manipulations also caused significant delays in sporulation. A separate study showed that the overexpression of another EBP gene, *ERG10*, caused reduction in sporulation rates [50]. Two of three *ERG11* isoform overexpression strains showed increased overall ergosterol content of the cell, as well as increased sporulation [51]. Thus, it appears that the right amount of ergosterol and activities of the EBP enzymes are critical in sporulation.

Here, we showed that deletion of *SRE1*, a gene that encodes a transcription factor responsible for increased EBP pathway gene expression under stress conditions such as hypoxia, nearly abolishes sporulation in unilateral crosses in *C. neoformans*. Furthermore, the overexpression of some EBP genes, such as *ERG2, ERG11*, and *ERG25*, can partially rescue the sporulation defect of the *sre1*Δ mutant. Thus, enhanced ergosterol biosynthesis is required for successful sporogenesis of the *sre1*Δ mutant. Surprisingly, the hypoxia and ergosterol regulator *SrbA* gene knockout in *A. fumigatus* does not have any notable defects in filamentation or conidiation compared to the wild type [52]. This could be attributed to the fact that *A. fumigatus* encodes an additional gene *SrbB*, which creates a protein similar to SrbA. SrbB influences the transcription of ergosterol biosynthesis but also has functions independent of SrbA [53].

One aspect that we found fascinating was that the overexpression of a single EBP gene was capable of conferring resistance to fluconazole in the wild-type background. This is surprising since the EBP pathway requires 23 enzymes in *C. neoformans.* The observation that even the overexpression of genes downstream of Erg11, the target of fluconazole, is capable of this effect is remarkable. Reducing the function of Erg11 via fluconazole should have a negative impact on biosynthetic steps downstream of this enzyme. Perhaps lanosterol, the substrate of Erg11, could serve (maybe poorly) as the substrate of the other downstream EBP enzymes, which allows the cells to produce other sterols that are functional in the presence of fluconazole in these overexpression strains. A sterol analysis of these overexpression strains in the presence and the absence of fluconazole would be necessary to determine if this was true.

In summary, we demonstrated here that ergosterol is necessary for sporogenesis in *C. neoformans*. Similar to oogenesis in high eukaryotes, basidia must accumulate a higher level of sterols by increasing the EBP pathway before initiating the production of spores.

## Figures and Tables

**Figure 1 jof-10-00106-f001:**
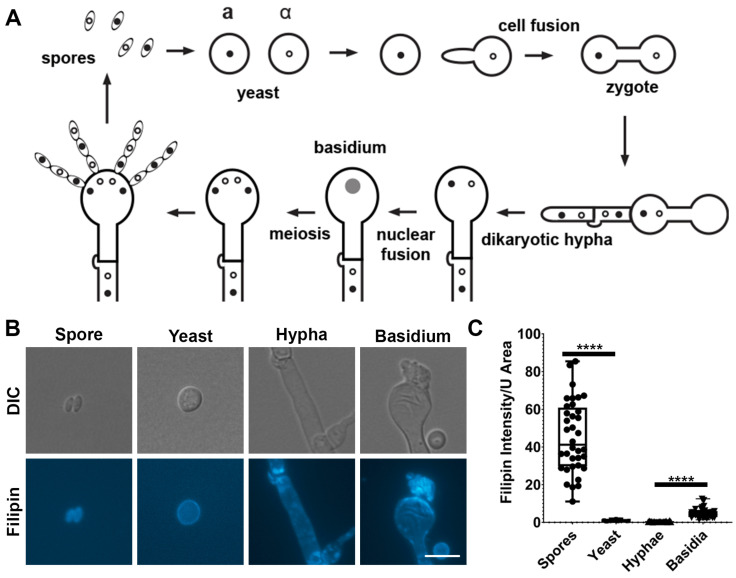
Ergosterol is enriched in spores and basidia based on filipin staining. (**A**) Diagram of bisexual reproduction of *C. neoformans*. (**B**) Mating mixtures were spotted onto V8 pH = 5 agar plates and incubated at 21 °C in the dark for two weeks. Spores and basidia were scooped and suspended into dH_2_O. Then, 0.5 mg/mL of filipin was used to stain the cells for 10 min before visualization. Scale bar = 10 µm. (**C**) Filipin fluorescence signal was quantified in Zen Pro software and standardized using approximate cell surface area (SA). The approximate SA of yeasts and basidia was calculated via the SA of a sphere, and the SA of hyphae and spores was calculated via the SA of a cylinder (see Materials and Methods for equations). Cell diameters and lengths were also measured in Zen Pro. Kruskal–Wallace test was used to assess statistical significance. **** *p* < 0.0001.

**Figure 2 jof-10-00106-f002:**
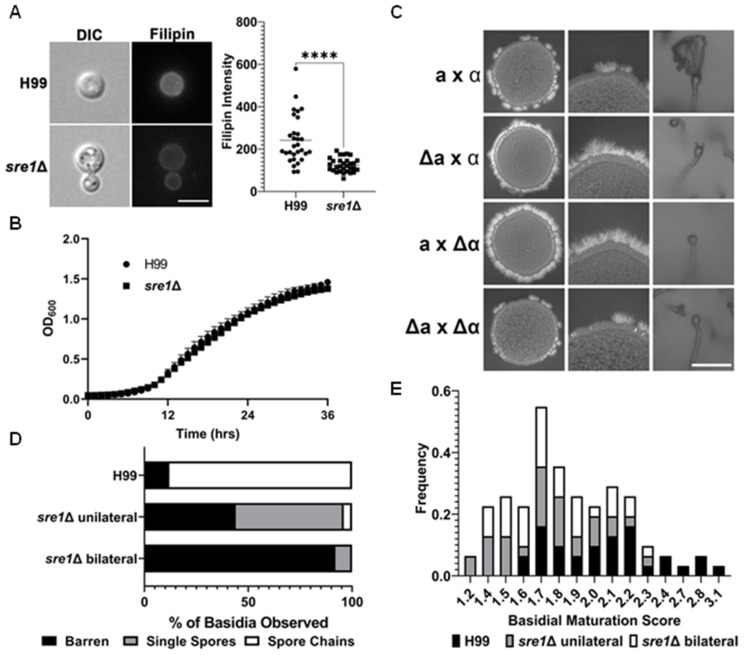
*sre1*Δ has a sporulation defect in both unilateral and bilateral crosses. (**A**) H99 and *sre1*Δ cells were grown to exponential growth in YPD media and stained with filipin for 10 min as in Figure 1 before visualization. Quantification of filipin fluorescent intensity was performed in Zen Pro. Scale bar = 10 µm. Mann–Whitney was used to assess statistical significance. **** *p* < 0.0001. (**B**) Cells were inoculated into a 96 well plate with a starting OD_600_ = 0.1 and shaken at 30 °C for 36 h. OD_600_ was read every hour. (**C**) Wild type (H99α x KN99**a**), *sre1*Δ unilateral, and *sre1*Δ bilateral crosses were made by mixing equal numbers of cells and spotting the mixed cells onto V8 pH = 5 agar plates. Plates were incubated at 21 °C in the dark for 20 days. The basidia and associated spore chains were imaged from the plate under microscope with a 20× objective. Scale bar = 100 µm. (**D**) 100 basidia from colonies in Panel C were visualized via light microscopy and imaged. Basidia were classified based on if they lacked spores (barren), had single/abnormal spore numbers, or had four spores/chains. (**E**) Basidia and hyphal diameters from the crosses in Panel C were measured in the Zen Pro software. The basidial maturation score was calculated per the inset figure.

**Figure 3 jof-10-00106-f003:**
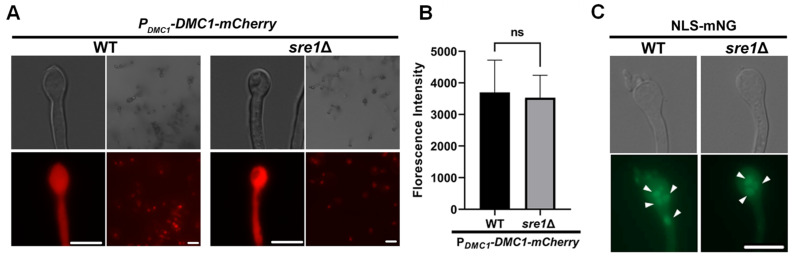
Meiosis is occurring in the basidia generated by *sre1*Δ crosses. (**A**) A strain with Dmc1-mCherry fusion protein with its expression driven by the native *DMC1* promoter was mated with wild-type KN99**a** cells. A *sre1*Δ strain harboring the Dmc1-mCherry fusion protein was also mated with KN99**a**. Mating crosses on V8 pH = 5 agar plates were incubated at 21 °C in the dark for two weeks. The plates were examined directly for basidia and Dmc1 expression, and basidia were scraped from the plate and placed onto agarose slides for microscopic examination. Scale bar = 10 µm. (**B**) Fluorescence intensity of the basidia from panel A was quantified using Zen Pro. Error bars reflect standard deviation. Student’s *t*-test was used to assess statistical significance. ns = not significant. (**C**) A strain constitutively expressing mNeonGreen fused with a nuclear localization signal (NLS-mNG) was mated with KN99**a** or the *sre1*Δ**a** mutant. After mating on V8 agar in the dark for two weeks, basidia were scraped as in panel A and visualized for nuclei, indicated by white arrows. Scale bar = 10 µm.

**Figure 4 jof-10-00106-f004:**
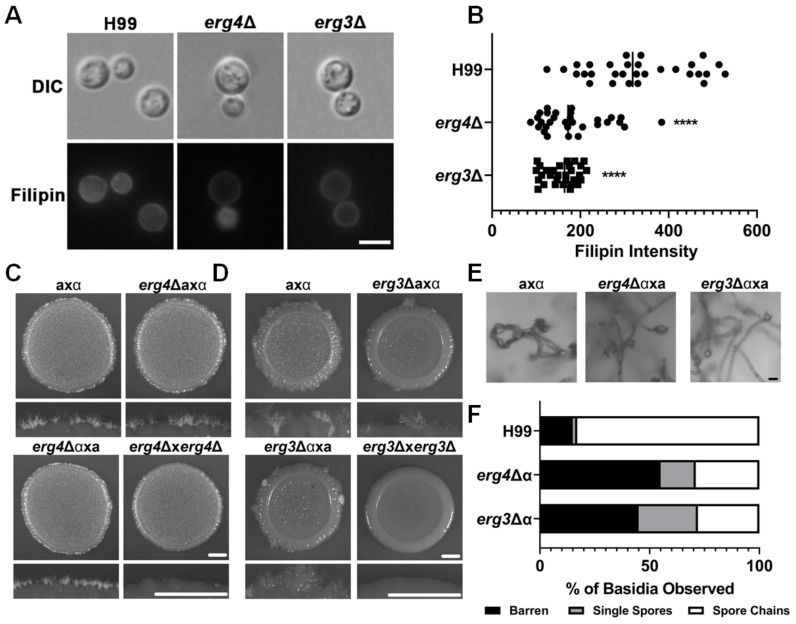
Deletion of EBP genes negatively impacts mating. (**A**) H99, *erg3*Δ, and *erg4*Δ cells were grown to exponential growth in YPD media and stained with filipin for 10 min as in Figure 1 before visualization. (**B**) Quantification of filipin fluorescent intensity was performed in Zen Pro. Scale bar = 5 µm. Mann–Whitney was used to assess statistical significance **** = <0.0001. (**C**) Serotype A crosses (wild type, *erg4*Δ unilateral, and *erg4*Δ bilateral) were spotted onto V8 pH = 5 agar and incubated at 21 °C in the dark for 10 days. The whole colonies and colony edges were visualized via a stereoscope. Scale bar = 100 µm. (**D**) *erg3*Δ crosses were set up in the same way as in panel A. The whole colonies and colony edges were visualized. (**E**) Representative images of basidia from the *erg4*Δ and *erg3*Δ unilateral crosses. Basidia from the *erg4*Δ unilateral cross were visualized at 10x objective via a light microscope. Scale bar = 5 µm. (**F**) Quantification of the types of basidia for the unilateral and wild-type crosses as in Figure 2D.

**Figure 5 jof-10-00106-f005:**
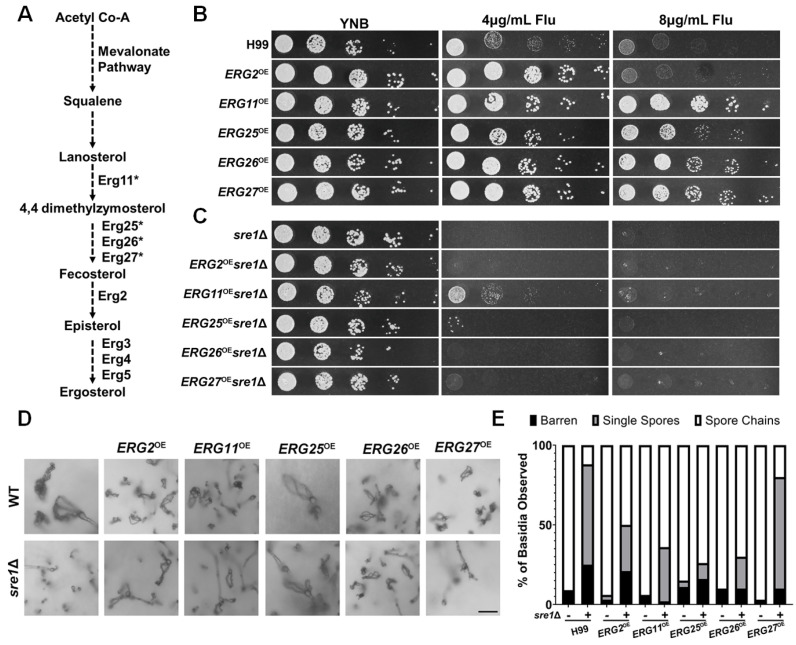
Overexpressed ergosterol biosynthetic genes are functional and can restore successful sporulation in the *sre1*Δ mutant. (**A**) Abbreviated ergosterol biosynthesis pathway highlighting ergosterol genes of relevance for this report. Essential Erg enzymes are marked with an asterisk (*). (**B**) The indicated EBP gene overexpression strains in the H99 were serially diluted and spotted onto YNB and YNB+fluconazole agar media. (**C**) EBP overexpression strains in the *sre1*Δ mutant background were spotted as in panel B. Plates were incubated at 30 °C for two days before imaging. (**D**) The indicated EBP gene overexpression strains with or without the *SRE1* gene were crossed unilaterally with the wildtype strain of the opposite mating type. After two weeks, basidia were visualized to determine sporulation frequencies. Scale bar = 10 µm. (**E**) Frequency of basidia with abnormal spores and spore chains for the crosses tested in Panel A.

## Data Availability

The data presented in this study are available herein and in the Appendix A.

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
