# Peer review of "Ergosterol Is Critical for Sporogenesis in Cryptococcus neoformans"

_jof, 2024, doi:10.3390/jof10020106_

Round 1

Reviewer 1 Report

Comments and Suggestions for Authors

This manuscript describes a role of ergosterol in the completion of the sexual cycle in cryptococci.

The experiments were well designed and execution, and I have little to comment on matters of content.

A few comments are in order on matters of data analysis and presentation. To wit:

General comment about statistical analyses: nowhere do you say what the error bars in your histograms are. SD? CI? I hope they're not SEMs, as these aren't a measure of scatter.

Section 2.6: please indicate the objective models used, including the numerical aperture, in the microscopy experiments.

Figure 1B/C and related text: I'm not sure FI/SA is the best way to compare sterol abundance among cell types. Figure 1B shows that while hyphae are poor in ergosterol, it's abundant around septa, and it also concentrates closer to the tip of basidia. Of course, your quantification data in figure 1C show that by your chosen measure, ergosterol is more abundant in spores indeed, but the micrographs show plain abundance isn't the entire story.

I suggest using Kruskal-Wallis instead of Student's for this experiment. You can't assume a normal distribution for your data, and you're comparing more than two experimental samples.

Figure 2 and related test: again, Mann-Whitney is better than Student's in figure 2A, as normality can't be assumed.

Figure 2B is a bit of a head-scratcher. Why did you choose spot dilution instead of a standard growth curve by optical density? Unless your mutant clumps or otherwise grows in a non-uniform way, optical density is the golden standard for the growth of planktonic microorganisms, whereas a spot dilution only provides snapshots of growth at arbitrarily chosen moments. Furthermore, growth curves allow for statistical comparison. I understand you're merely replicating the observations from Jung 2015, but that study used spot dilutions because it was comparing the impact of antifungals, so the controls were done in the same fashion. If you don't have any new data to add, just remove figure 2B altogether and state in the text that you replicated the growth profile of the mutant in your hands (it's the same one as in Jung 2015, anyway).

On figure 2D, you don't state how many basidia per group were assessed for the presence of spore chains. This is even more important because you didn't perform statistics to validate the observed differences. This is okay if the number of basidia counted is high, because then the data speak for themselves, but if you counted just a few basidia, then we need statistics even when the differences observed are high. This is also important for figure 6D.

Figure 4 and related text: again, Mann-Whitney is better than Student's in figures 4C and 4E.

Figure 5 and related text:

On line 373, you probably mean figure 6A, not 5A.

Please indicate a reference for the information on non-essential EBP enzymes in line 374.

On figure 5D, please indicate which mating type was the mutant in the crosses used for spore chain counting.

I probably shouldn't carp about supplemental figures, but for reference, the comparison of RT-PCR data on figure S2 shouldn't use the fold-change values, since these are mathematical derivations of the things actually measured, deltaC(t)s. It's possible to perform statistical analysis on them, but it's more complicated as it requires dealing with geometric means etc. It's better to remake the histograms to show the deltaC(s)s instead of fold-changes, which makes it clearer what the error bars are (you don't say, but I'm assuming they're SDs). Then you can compare them using Student's or Mann-Whitney (better), and if you want to show the fold change, just place it numerically above the histogram of the mutant.

Comments on the Quality of English Language

The manuscript is very well-written, but a close reading by a rested mind can spot a few mistakes, like:

12: "is enriched", not "are enriched"

17: "preceding"

These are of small importance, but you may give the final version an extra polish if you have time.

Reviewer 2 Report

Comments and Suggestions for Authors

Matha and colleagues present an article on aspects of the role of the ergosterol in the processes of sporogenesis in Cryptococcus neoformans, highlighting the roles of the transcription factor Sre1 and individual genes of the biosynthetic pathway. In general, the article is comprehensibly written and includes interesting information for people working with this fungus in this field. Below are some comments for the authors’ consideration.

In my opinion, authors should restructure the introduction, focusing more on ergosterol and pathway homologues in other fungi and consider moving the parts that are relevant to the life-cycle of C. neoformans in the first chapter of the results, where also Figure 1A is shown.

Line 67: References 21 and 22 do not seem appropriate in this context.

Does the overexpression of Sre1 lead to any phenotype?

Figure 2A: Based on the image shown in 2A, deletion of SRE1 not only reduces the fillipin staining, but totally inhibits staining. Is this correct? Perhaps another image would be more appropriate for sre1Δ? Including a bright-field image would be helpful in any case.

Figure 4: Please explain the use of Mitotracker. What is the correlation and relevance here?

Line 373: Figure 5A has nothing to do with “…series of reactions to transform acetyl Co-A into ergosterol”.

Figure 5: A better explanation is necessary for the phenotypes related to the growth tests. Please also include a wt in 5c and 5d.

Please consider including and discussing fillipin staining or mNG fluorescence in the single erg-gene deletions.

It would be important and most interesting to know if exogenous supplementation of the pathway intermediates are suppressing the observed phenotypes.

Line 402: Which EBP genes are upregulated by Sre1?

Lines 414-422. Please consider restructuring and better explaining this paragraph of the results on fluconazole.

Some more information and speculation could be added in the overall discussion. In addition, interesting studies on genes involved in ergosterol synthesis have also been conducted in Aspergillus species other than A. fumigatus and all these articles should also be cited and discussed.

Round 2

Reviewer 2 Report

Comments and Suggestions for Authors

I thank the authors for considering these suggestions. The revised manuscript has been improved. Concerning the discussion of relevant results in other Aspergillus species, I would also like to draw the authors’ attention on articles of A. nidulans (like for example https://doi.org/10.3390/jof7070514), which may be of interest. I have no further comments for the authors.